# Factors Associated with Prenatal Smoking Cessation Interventions among Public Health Nurses in Japan

**DOI:** 10.3390/ijerph17176135

**Published:** 2020-08-24

**Authors:** Meng Li, Reiko Okamoto, Aoki Tada, Misaki Kiya

**Affiliations:** Division of Health Sciences, Osaka University Graduate School of Medicine, Suita City 565-0871, Osaka Prefecture, Japan; 25b18817@sahs.med.osaka-u.ac.jp (M.L.); aotada@sahs.med.osaka-u.ac.jp (A.T.); misaki_kiya@sahs.med.osaka-u.ac.jp (M.K.)

**Keywords:** smoking cessation, pregnant women, psychological intervention, nurses, Japan

## Abstract

This study aimed to identify the factors associated with prenatal smoking cessation interventions based on the 5As model among public health nurses (PHNs) in Japan. A nationwide cross-sectional study was conducted from December 2019 to February 2020 via a self-administered questionnaire. The study subjects were 1988 PHNs working in 431 health centers of municipalities and special wards across the country. Of the 1988 questionnaires mailed, 521 responses (26.2%) were included in the analysis. Of the 521 responses, most of the respondents were female (98.1%) and the mean age was 37.5 years. There were statistically significant differences on age, work regions, experience years working as a PHN and smoking cessation training after becoming a PHN in implementing the 5As. Self-efficacy, professional development competency, research utilization competency, age and experience years working as a PHN were positively associated with the 5As. Social nicotine dependence was negatively associated with the 5As. Furthermore, self-efficacy mediated the relationship between the 5As and professional development competency, research utilization competency, social nicotine dependence, age and experience years working as a PHN. In the future, smoking cessation intervention training should be widely implemented to improve self-efficacy and prenatal smoking cessation interventions among Japanese PHNs.

## 1. Introduction

Maternal smoking and second-hand smoke exposure during pregnancy are two modifiable risk factors for maternal and fetal health [1]. Many studies have reported that maternal smoking during pregnancy was associated with increased risks for ectopic pregnancy, preterm premature rupture of the membrane, placenta previa, abruption placenta, miscarriage, stillbirth, preterm birth (less than 37 weeks gestation), low birth weight (less than 2500 g), small for gestational age and congenital anomalies such as cleft lip [2,3,4,5,6,7,8,9,10]. After birth, the risk for sudden infant death syndrome is increased among the offspring of women who smoked during or after pregnancy [11]. Into childhood, offspring may experience increased risks of respiratory problems, cancers, neurodevelopmental and behavioral problems, as well as increased long-term risks of non-communicable diseases [12]. In addition, maternal exposure to second-hand smoke during pregnancy has been reported to be associated with spontaneous abortion, stillbirth, preterm birth, small for gestational age, low birth weight, neonatal asphyxia and neural tube defects [13,14,15,16,17,18].

According to a recent systematic review, the global prevalence of maternal smoking during pregnancy is estimated to be 1.7%. Twenty-nine of 174 countries have a prevalence of smoking during pregnancy greater than 10%, and 12 countries have a prevalence greater than 20% [19,20]. In high-income countries such as the USA, Denmark and Sweden, the prevalence of maternal smoking during pregnancy has declined from between 20% and 35% in the 1980s to between 10% and 20% in the 2000s and less than 10% in 2010. In contrast, in low- and middle-income countries, the prevalence is between 15% and 37%, and, in some of these countries, this problem does not seem to decrease [20,21]. Therefore, it is an urgent public health concern to protect pregnant women from the harmful effects of tobacco.

To encourage women to stop smoking during pregnancy, psychological intervention is recommended as the first line of treatment [22,23]. The World Health Organization also recommends that healthcare providers routinely offer advice and psychological interventions for tobacco cessation to all pregnant women who are either current tobacco users or recent tobacco quitters [1]. As a brief psychological intervention method, the 5As intervention model for smoking cessation (Ask, Advise, Assess, Assist and Arrange for follow-up) is recommended to be used at the first antenatal appointment and during subsequent appointments [1,24,25]. In addition, previous studies have reported that pregnancy was a good opportunity to quit smoking because patients visit the clinic regularly and may be motivated effectively [1,26]. Overall, psychological intervention should be prioritized to help women to stop smoking during pregnancy.

In Japan, the prevalence of maternal smoking during pregnancy is estimated to be between 5.1% and 7.5% [19], and the prevalence of paternal smoking is 47% for the first trimester and 46% for the second and third trimesters [27]. In light of the troubling fact, previous studies have reported that PHNs could make important contributions to reduce maternal smoking and second-hand smoke exposure during pregnancy and the postpartum period because they have a variety of opportunities to engage with mothers and children from pregnancy to childhood [28]. In addition, Japanese guidelines for smoking cessation recommend healthcare providers to adopt the 5As model for smoking cessation in the general practice settings such as outpatient care and medical examination [26]. However, there are few studies that focus on the prenatal smoking cessation interventions. Therefore, this study aimed to identify the factors associated with prenatal smoking cessation interventions based on the 5As model among Japanese PHNs.

## 2. Methods

### 2.1. Study Design

This is a nationwide cross-sectional study from December 2019 to February 2020 via a self-administered questionnaire.

### 2.2. Study Subjects and Ethical Approval 

The study subjects were 1988 PHNs working in 431 health centers of municipalities and special wards and they were randomly selected from Official Statistics of Japan in e-Stat [29]. In Japan, health centers are government agencies located in special wards or municipalities across the country. PHNs who work there are employed by that municipality as public health specialists, and they engage with community members with a wide range of health levels including infants, young children, elderly people, pregnant women and people with disabilities [30].

Before conducting the nationwide survey, we carried out semi-structured interviews with 10 PHNs who had an extensive working experience in mother and child health to obtain their opinions on prenatal smoking cessation interventions.

This study was approved by the Ethical Committee of Observation Research at the Osaka University Hospital (Approval No. 19308).

### 2.3. Data Collection and Measures

The self-administered questionnaire was developed based on the Integrated Change (I-Change) model. The I-Change model is an integrated model for explaining motivational and behavioral change. The model demonstrates that behavior is the result of a person’s intentions and abilities. Important abilities are plans to implement intentions by specific actions to reach the goal behavior and actual skills. The intention to perform a behavior is most proximally determined by the motivational factors such as attitudes, social influences and self-efficacy. Motivational factors are determined by various predisposing factors, information factors (the quality of messages, channels and sources used) and awareness factors (knowledge, risk perceptions and cues to action). In addition, the model demonstrates that predisposing factors may directly influence behavior or indirectly influence behavior through the motivational factors [31,32]. The design of previous studies based on the I-Change model have demonstrated its usefulness in explaining health professionals’ behaviors related to smoking cessation interventions [33,34,35].

In this study, the self-administered questionnaire included questions about prenatal smoking cessation interventions based on the 5As model, motivational factors, ability factors and barriers and predisposing factors (demographic–professional factors and smoking cessation intervention factors). 

#### 2.3.1. Development of Prenatal Smoking Cessation Intervention Items

Development of prenatal smoking cessation intervention items was divided into three stages. Firstly, we extracted 68 items from the previous qualitative study and a literature review, and then categorized these items based on the 5As model. The literature review included previous papers [23,24,25,36,37,38,39] and guideline books [1,26,40,41,42,43]. Secondly, we conducted a pretest of 27 PHNs to obtain some opinions on these items and wording, and then we carefully compared these items and corrected the wording to complete the item draft (Appendix A). This item draft included 36 items rated on a 6-point Likert scale (0 to 5 representing “not at all” to “always”), and it was used for the nationwide cross-sectional study. To increase the response rate, we sent a reminder about one month after the questionnaire was mailed out. Finally, 30 items were effectively confirmed on prenatal smoking cessation interventions based on item analysis (Appendix A) and internal consistency reliability (Cronbach’s alpha = 0.928). The scores of the 30 items range from 0 to 150, and higher total scores indicate higher levels of implementing prenatal smoking cessation interventions. 

#### 2.3.2. Motivational Factors 

Attitude was measured by the Kano Test for Social Nicotine Dependence (KTSND), which is a useful method to evaluate psychological aspects of smoking. It is not only used to reflect the smoking status and the stages for quitting smoking, but also reflect the affinity for smoking among non-smokers. In this study, the KTSND showed an acceptable internal consistency reliability (Cronbach’s alpha = 0.809). The KTSND has 10 items with a total score of 30, and higher total scores indicate higher levels in the affinity for smoking [44,45].

Social influence was measured by the professional development scale (PDS) that reflects professional development competency as a PHN. It includes 16 self-report items rated on a 6-point Likert scale (0 to 5 representing “not at all” to “always”). In this study, the PDS showed a relatively high internal consistency reliability (Cronbach’s alpha = 0.910). The scores of PDS range from 0 to 80, and higher total scores indicate higher levels of professional development competency as a PHN [46]. 

Self-efficacy was measured by self-efficacy scale (SE), which was used to measure nurses’ confidence for smoking cessation intervention in general hospitals. It includes 12 self-report items rated on a 7-point Likert scale (1 to 7 representing “not confident at all” to “very confident”). In this study, the self-efficacy scale showed a relatively high internal consistency reliability (Cronbach’s alpha = 0.915). The scores of the self-efficacy scale range from 12 to 84, and higher total scores indicate higher levels of self-efficacy in implementing smoking cessation intervention [47]. 

#### 2.3.3. Ability Factors and Barriers

Ability factors and barriers were measured by the research utilization competency (RUC) scale, which is used to quantify research results utilization competency of PHNs. It includes 10 self-report items rated on a 6-point Likert scale (1 to 6 representing “Strongly disagree” to “Strongly agree”). In this study, the RUC scale showed a relatively high internal consistency reliability (Cronbach’s alpha = 0.914). The scores of RUC scale range from 10 to 60, and higher total scores indicate higher levels of research results utilization competency [48].

#### 2.3.4. Predisposing Factors

For the predisposing factors, demographic–professional factors included gender, age, experience years working as a PHN, work position, work region, PHNs and cohabitation’s smoking behavior and educational background. Smoking cessation intervention factors included the experience of smoking cessation training in college and after becoming a PHN. 

### 2.4. Data Analysis 

Statistical analysis was divided into four stages. Firstly, gender, age, experience years working as a PHN, work position, work region, PHNs and cohabitation’s smoking behavior, educational background, the experience of smoking cessation training in college and after becoming a PHN were presented by frequency distribution and percentage; while age and experience years working as a PHN were also presented as mean and standard deviation (SD). Secondly, independent samples *t* test and one-way analysis of variance (ANOVA) were used to assess the difference of implementing the 5As by predisposing factors. Thirdly, Pearson’s correlation was used to examine the relationships among the 5As, self-efficacy, PDS, RUC, KTSND, age and experience years working as a PHN. Finally, path analysis was used to construct the equation of prenatal smoking cessation intervention model and the goodness-of-fit of the model was evaluated with the goodness-of-fit index (GFI), adjusted goodness-of-fit index (AGFI), normed fit index (NFI), comparative fit index (CFI) and root mean square error of approximation (RMSEA).

Statistical analysis was performed using SPSS Statistics 25.0 (IBM Japan, Tokyo, Japan) and SPSS Amos 25.0 (IBM Japan, Tokyo, Japan) Graphics for Microsoft Windows. For all analyses, *p* < 0.05 was considered statistically significant. 

## 3. Results

Of the 1988 questionnaires mailed, 604 PHNs responded to the survey (30.4%). Eighty-three responses were excluded because they missed an answer or had multiple answers to a question. Finally, 521 responses (26.2%) were included in the analysis.

### 3.1. Characteristics of the Respondents Included in the Analysis

Characteristics of the respondent included in the analysis are shown in Table 1. Most of the respondents were female (98.1%) and the mean age was 37.5 (±9.35) years. The mean experience years working as a PHN were 12.4 (±9.48). More than half of the respondents (59.7%) had staff position, followed by assistant managerial level position (33.9%). Respondents from Kanto/Koshinetsu accounted for the largest share of the sample (25.7%), followed by those from Tokai/Hokkiku (20.5%) and Hokkaido/Tohoku (20.0%). Almost all of the respondents (99.6%) did not smoke and the majority of cohabitation (84.6%) did not smoke. Approximately half of the respondents (48.6%) reported the experience of smoking cessation intervention training in college and more than half of respondents (56.0%) reported the experience of smoking cessation intervention training after becoming a PHN. More than half of the respondents (56.0%) completed a four-year university, followed by those who completed vocational college (27.6%) and junior college (15.2%). 

### 3.2. Difference of Implementing the 5As by Predisposing Factors

Difference of implementing the 5As by predisposing factors are shown in Table 2. PHNs aged 40–49 were more likely to implement the 5As and provide advice and assistance than PHNs aged less than 29 years. PHNs aged 40–49 were more likely to implement the 5As and provide assistance than PHNs aged 30–39. PHNs who had 16–25 experience years working as a PHN were more likely to implement the 5As and provide advice and assistance than PHNs who had less than five years of experience and PHNs who had 6–15 years. PHNs who had more than 26 experience years were more likely to provide assistance than PHNs who had less than five years of experience. In work position, PHNs with assistant managerial level position was more likely to provide advice and assistance than staff position. In work region, PHNs who were from Chugoku/Shikoku were less likely to implement the 5As and provide advice and assistance than PHNs who were from Hokkaido/Tohoku, Kanto/Koshinetsu, Tokai/Hokkiku and Kinki. PHNs who were from Chugoku/Shikoku were less likely to provide advice than PHNs who were from Kyushu/Okinawa. PHNs who were from Chugoku/Shikoku were less likely to provide arrangement than PHNs who were from Kinki. PHNs who or whose cohabitation were smokers were more likely to provide assistance than PHNs who or whose cohabitation were non-smokers. PHNs who had the experience of smoking cessation training after becoming a PHN were more likely to implement the 5As and provide advice and assistance than PHNs who did not have the experience. PHNs whose educational background was vocational college were more likely to provide assistance than PHNs whose educational background was above university.

### 3.3. The Relationships among Study Variables

The relationships among study variables are shown in Table 3. SE (r = 0.621, *p* < 0.01), PDS (r = 0.331, *p* < 0.01), RUC (r = 0.413, *p* < 0.01), age (r = 0.125, *p* < 0.01) and experience years (r = 0.136, *p* < 0.01) were positively associated with the 5As. PDS (r = 0.369, *p* < 0.01), RUC (r = 0.418, *p* < 0.01), age (r = 0.256, *p* < 0.01) and experience years (r = 0.274, *p* < 0.01) were positively associated with SE. In addition, KTSND was negatively associated with the 5As (r = −0.113, *p* < 0.05) and SE (r = −0.109, *p* < 0.05).

### 3.4. Construction of the Prenatal Smoking Cessation Intervention Model

Variables significantly associated with the 5As were used to construct the prenatal smoking cessation intervention model (Figure 1). The results of the prenatal smoking cessation intervention model were the following: Chi-square of 13.372 and statistical degrees of freedom of 9 (*p* = 0.146). GFI was 0.993, AGFI was 0.977, NFI was 0.991, CFI was 0.997 and RMSEA was 0.031, which indicated acceptable goodness-of-fit.

From the model, SE (β = 0.52, *p* < 0.001) was directly and positively associated with the 5As and mediated the relationship between the 5As and RUC, PDS, KTSND, age and experience years. The standardized beta for the direct path from RUC to the 5As was 0.17 (*p* < 0.001), indicating partial mediation of SE. The standardized beta for the direct path from PDS to the 5As was 0.05 (*p* > 0.05), indicating partial mediation of SE. The standardized beta for the direct path from KTSND to the 5As was −0.06 (*p* > 0.05), indicating partial mediation of SE. In addition, RUC (β = 0.31, *p* < 0.001) and PDS (β = 0.20, *p* < 0.001) were positively associated with SE, while KTSND (β = −0.079, *p* < 0.05) was negatively associated with SE. 

## 4. Discussion

To the best of our knowledge, this is the first study to identify the factors associated with prenatal smoking cessation interventions based on the 5As model among Japanese PHNs. In this nationwide cross-sectional study, we selected PHNs working in health centers of municipalities and special wards as the study subjects because they have a variety of opportunities to engage with mothers and children from pregnancy to childhood [28]. The mean age of PHNs in this sample was 37.5 years, with the largest share of participants in their 30s (31.7%), followed by their 40s (30.5%). The result is almost consistent with PHNs activity area survey conducted by the Ministry of Health, Labor and Welfare, which reported that most people were in their 40s (30.2%), followed by people in their 30s (29.7%) [49]. In addition, this sample covered almost all work regions across the country from 9.8% in Kyushu/Okinawa to 25.7% in Kanto/Koshinetsu. Therefore, this sample can be confirmed as representing PHNs across the country. 

In this study, there are several findings that need to be discussed. Firstly, we found that age was positively associated with the 5As. PHNs aged 40–49 were more likely to implement the 5As and provide advice and assistance than PHNs aged less than 29 years. PHNs aged 40–49 were more likely to implement the 5As and provide assistance than PHNs aged 30–39. Wetta-Hall et al. (2005) reported that older office-based nurses aged 41–50 were more likely to assess tobacco use and younger middle-aged nurses aged 31–40 were more likely to assess a patient’s interest in stopping tobacco use in Kansas, USA [50]. Price et al. (2006) reported that there was a significant but modest positive correlation between the use of the 5As model and older age of nurse-midwives in Ohio, USA. Mak et al. (2018) reported that more mature nurses were more likely to participate in providing advice and assistance and in making arrangements in Hong Kong [51]. In addition, Sarna et al. (2001) reported that younger age was associated with the greatest barriers to delivering tobacco interventions among nurse members of the Oncology Nursing Society in the USA [52]. However, Leung et al. (2009) reported that nurses aged 26–30 were less likely than nurses aged 21–25 to offer “follow-through” activities in the Hong Kong sample, while nurses aged 31–35 were less likely than nurses aged 21–25 to implement “follow-through” activities in the Guangzhou sample [53]. Overall, mature nurses were more likely to implement the smoking cessation interventions. 

Secondly, our study revealed that experience years as a PHN were positively associated with the 5As. PHNs who had 16–25 experience years were more likely to implement the 5As and provide advice and assistance than PHNs who had less than 15 years of experience. PHNs who had more than 26 years were more likely to provide assistance than PHNs who had less than five years of experience. In two previous studies, Wetta-Hall et al. (2005) reported that office-based nurses were less likely to assess patient tobacco use if they had practiced as a nurse for less than or equal to five years [50]. Sarna et al. (2012) reported that nurses with less than 15 years of practice were more likely to consistently Ask, Advise, Assess and Assist smokers to quit, but not Arrange or Refer to a Quitline in the USA [54]. Overall, nurses with more experience years were more likely to implement the smoking cessation interventions. Furthermore, the two results imply that mature nurses may have more work experience to use the 5As and then the experience motivates them to implement the 5As, although these studies do not provide the direct association between age and experience years.

Thirdly, our study revealed that PHNs who had the experience of smoking cessation training after becoming a PHN were more likely to implement the 5As and provide advice and assistance than PHNs who did not have the experience. Wetta-Hall et al. (2005) reported that office-based nurses with tobacco-related continuing education in the previous year more likely to assess patient tobacco use, to assess a patient’s interest in stopping tobacco use and to give tobacco cessation advice [50]. Several other previous studies reported that nurses with prior training were more likely to implement the 5As [55,56,57]. In addition, nurses with prior training were more positive about their role in providing cessation interventions and they were more competent in performing tasks related to smoking cessation interventions; trained nurses also were significantly more aware of factors positively and negatively influencing cessation interventions in China [55]. Moreover, practice nurses with training were more likely to report recommending nicotine replacement therapy (NRT), believe that NRT is effective enough to justify its cost and believe that NRT should be on general sale in the UK [58]. 

Fourthly, our study also revealed that PHNs who were from Chugoku/Shikoku were less likely to implement the 5As, however PHNs who were from Kinki and Tokai/Hokkiku were more likely to implement the 5As. In Japan, there are several smoking cessation intervention training programs such as Japan Smoking Cessation Training Outreach Project (J-STOP) that provide the training contents including the smoking-related knowledge, behavioral counselling and medication treatment. We think that PHNs who were from the work region where more PHNs complete the training programs may be more likely to implement the 5As. Therefore, this consideration may suggest that it is necessary to promote more PHNs who were from Chugoku/Shikoku to complete smoking cessation intervention training programs. 

Fifthly, our study revealed that self-efficacy was positively associated with the 5As. This result is consistent with several previous studies which reported that nurses or nurse-midwives’ self-efficacy was positively associated with smoking cessation interventions [51,56,59,60,61]. Furthermore, our study revealed that self-efficacy mediated the relationship between the 5As and professional development competency, research utilization competency, social nicotine dependence, age and experience years working as a PHN. The results are consistent with a previous study that reported that self-efficacy was positively related to providing smoking cessation services and mediated tobacco control policy, employer support, smoking cessation services training and attitude toward smoking cessation services that predicted provision of smoking cessation services [56]. 

Regarding the strategies to improve healthcare professionals’ self-efficacy and smoking cessation interventions, the meta-analyses by Cochrane Collaboration reported that healthcare professionals who had received training were more likely to perform tasks of smoking cessation than compared with their untrained control counterparts [62]. Moreover, another previous study in Malaysia reported that an 8-h smoking cessation intervention training program including interactive lectures, practical sessions and role-play sessions could improve healthcare providers’ knowledge, attitude and self-efficacy on the smoking cessation interventions. In this study, the interactive lectures consist of introduction, tobacco control policy, national strategic plan, harm to health, smoking as an addiction behavior and pharmacological and behavioral therapy on smoking cessation. The practical session consists of assessment on how to use the tobacco dependence instrument, how to monitor carbon monoxide levels using Smokerlyzer and how to run the quit smoking clinic. The role-play was based on the 5As counselling approach where the participants acted as smoking cessation providers and the facilitator acted as a patient [63]. Overall, smoking cessation intervention training was an evidence-based strategy to improve healthcare professionals’ self-efficacy and smoking cessation intervention.

Finally, several previous studies reported that smokers had more significant social nicotine dependence than non-smokers and ex-smokers [44,64,65,66], and smokers with higher KTSND scores had difficulty beginning to quit smoking among the stages for quitting smoking [44]. Another previous study reported that non-smoking and ex-smoking teachers with high KTSND scores might underestimate the health hazards of second-hand smoke and overestimate the recommended protective measures [67]. In this study, we found that the KTSND was negatively associated with the 5As. This result is consistent with one previous study which reported that participants in the Annual Meeting of the Japan Lung Cancer Society who answered that they are not interested in smoking cessation intervention showed significantly higher KTSND scores [64]. Overall, health workers with significant social nicotine dependence were less likely to implement the smoking cessation interventions. As an ability factor, our study firstly revealed that research utilization competency of PHNs was positively associated with prenatal smoking cessation interventions based on the 5As model. A previous study including 1054 primary healthcare nurses in Sweden reported that the ability and research use were significant determinants of attitudes towards research and use of research findings [68]. The two results imply that the research utilization competency may firstly have an influence on attitude toward the use of research findings and then on the smoking cessation interventions. Overall, nurses with higher research utilization competency were more likely to implement the smoking cessation interventions.

In the future, there are two suggestions for the prenatal smoking cessation intervention services. On the one hand, smoking cessation intervention training should be widely implemented to improve self-efficacy and prenatal smoking cessation interventions among Japanese PHNs. On the other hand, self-efficacy can be further improved by promoting the research utilization competency, regulating the health professionals’ norms and changing the attitudes on smoking. 

The limitations of this study include sampling and response rates. PHNs in the sample are from Japan, thus the results may not be applicable to PHNs in other countries. Although PHNs in this study were randomly picked across the country, the effective response rate was only 26.2% and it may not represent all Japanese PHNs. 

## 5. Conclusions

In this study, we initially identified the factors associated with prenatal smoking cessation interventions based on the 5As model among Japanese PHNs. Age, experience years working as a PHN, work region, smoking cessation training after becoming a PHN, self-efficacy, professional development competency, research utilization competency and social nicotine dependence were found to be associated with prenatal smoking cessation interventions. In the future, smoking cessation intervention training should be widely implemented to improve self-efficacy and prenatal smoking cessation interventions among Japanese PHNs. 

## Figures and Tables

**Figure 1 ijerph-17-06135-f001:**
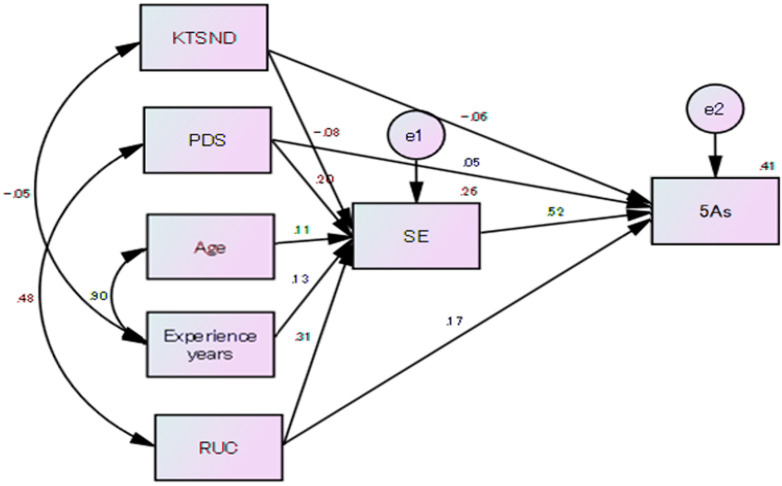
Prenatal smoking cessation intervention model (*n* = 521).

**Table 1 ijerph-17-06135-t001:** Characteristics of respondents included in the analysis (*n* = 521).

Characteristics	*n*	%	Mean (±SD)
Gender			
Female	511	98.1	
Male	10	1.9	
Age (years)			37.5 (±9.35)
≤29	134	25.7	
30–39	165	31.7	
40–49	159	30.5	
≥50	63	12.1	
Experience years working as a PHN (years)			12.4 (±9.48)
≤5	176	33.8	
6–15	156	29.9	
16–25	131	25.1	
≥26	58	11.1	
Work position			
Staff	311	59.7	
Assistant managerial level	177	33.9	
Managerial level	33	6.4	
Work region			
Hokkaido/Tohoku	104	20.0	
Kanto/Koshinetsu	134	25.7	
Tokai/Hokkiku	107	20.5	
Kinki	71	13.6	
Chugoku/Shikoku	54	10.4	
Kyushu/Okinawa	51	9.8	
PHNs’ smoking behavior			
No	519	99.6	
Yes	2	0.4	
Cohabitation’s smoking behavior			
No	441	84.6	
Yes	80	15.4	
Experience of smoking cessation intervention training in college			
No	268	51.4	
Yes	253	48.6	
Experience of smoking cessation intervention training after becoming a PHN			
No	229	44.0	
Yes	292	56.0	
Educational background			
Vocational college	144	27.6	
Junior college	79	15.2	
University (4 years)	292	56.0	
Master program	6	1.2	

PHN, public health nurse.

**Table 2 ijerph-17-06135-t002:** Difference of implementing the 5As by predisposing factors (*n* = 521).

Items	5As	Ask	Advise	Assess	Assist	Arrange
	Mean ± SD *p*	Mean ± SD *p*	Mean ± SD *p*	Mean ± SD *p*	Mean ± SD *p*	Mean ± SD *p*
Gender ^a^						
Female	99.2 ± 21.91	21.8 ± 3.28	46.6 ± 10.17	8.6 ± 4.31	12.9 ± 5.20	9.3 ± 4.31
Male	98.0 ± 26.81	21.1 ± 1.60	99.2 ± 21.99	10.5 ± 3.87	13.9 ± 6.15	8.3 ± 5.31
Age ^b^						
≤29	95.7 ± 21.95 *	21.5 ± 3.34	44.0 ± 10.37 **	8.6 ± 4.43	12.1 ± 4.96 **^,^***	9.5 ± 4.14
30–39	96.9 ± 21.31 *	21.7 ± 3.35	46.2 ± 9.91 **	8.2 ± 4.23	12.0 ± 4.86 ***	8.8 ± 4.39
40–49	103.6 ± 21.1 *	22.3 ± 2.81	48.6 ± 9.66 **	8.9 ± 4.30	14.3 ± 5.43 ***	9.6 ± 4.39
≥50	101.3 ± 24.2	21.6 ± 3.80	47.9 ± 10.86	9.2 ± 4.16	13.7 ± 5.36	8.9 ± 4.40
Experience years working as a PHN						
≤5	95.8 ± 22.74 **	21.5 ± 3.59	44.7 ± 10.70 **	8.6 ± 4.47	11.7 ± 5.04 **^,^***	9.3 ± 4.26
6–15	97.4 ± 20.73 *^,^**	21.6 ± 2.97	46.0 ± 9.75 *^,^**	8.2 ± 4.20	12.6 ± 5.05 **^,^***	9.1 ± 4.23
16–25	104.4 ± 20.3 *^,^**	22.4 ± 2.66	49.1 ± 9.02 *^,^**	8.9 ± 4.23	14.5 ± 5.17 **^,^***	9.4 ± 4.52
≥26	102.6 ± 24.3	21.9 ± 4.00	48.4 ± 10.93	9.3 ± 4.21	14.2 ± 5.23 **	9.0 ± 4.48
Work position						
Staff	97.2 ± 21.30	21.6 ± 3.22	45.4 ± 10.14 *	8.5 ± 4.28	12.4 ± 4.97 *	9.3 ± 4.20
Assistant managerial level	101.6 ± 22.68	22.1 ± 3.27	48.1 ± 10.00 *	8.7 ± 4.50	13.6 ± 5.61 *	9.1 ± 4.60
Managerial level	104.2 ± 23.05	22.2 ± 3.58	49.3 ± 10.65	9.4 ± 3.38	14.2 ± 4.87	9.2 ± 4.24
Work region						
Hokkaido/Tohoku	99.3 ± 23.51 *	21.8 ± 3.30	46.5 ± 11.22 **	8.3 ± 4.51	13.0 ± 5.24 *	9.7 ± 4.16
Kanto/Koshinetsu	99.3 ± 22.37 *	21.4 ± 3.65	46.5 ± 10.58 **	9.2 ± 4.35	13.4 ± 5.14 *	8.9 ± 4.25
Tokai/Hokkiku	102.7 ± 20.31 *	22.4 ± 3.00	48.1 ± 8.88 *^,^**	9.0 ± 4.36	13.4 ± 5.35 *^,^**	9.7 ± 4.21
Kinki	103.3 ± 20.54 *^,^**	21.9 ± 3.07	48.6 ± 9.36 *^,^**^,^***	8.7 ± 3.92	13.8 ± 4.92 *^,^**	10.3 ± 3.71 *
Chugoku/Shikoku	87.9 ± 22.23 *^,^**	21.7 ± 3.23	40.4 ± 10.08 *^,^**^,^***	7.6 ± 3.79	10.5 ± 5.14 *^,^**	7.7 ± 4.71 *
Kyushu/Okinawa	97.5 ± 19.34	21.4 ± 2.79	47.3 ± 8.63 **	8.4 ± 4.58	12.3 ± 4.97	8.1 ± 4.90
PHNs or cohabitation’s smoking behavior						
No	98.7 ± 22.03	21.8 ± 3.23	46.4 ± 10.28	8.6 ± 4.28	12.7 ± 5.23 *	9.2 ± 4.33
Yes	102.0 ± 21.61	21.7 ± 3.41	47.6 ± 9.78	9.0 ± 4.42	14.1 ± 5.00 *	9.5 ± 4.35
Experience of smoking cessation training in college						
No	98.2 ± 23.05	21.8 ± 3.41	46.1 ± 10.90	8.5 ± 4.45	12.9 ± 5.47	9.0 ± 4.40
Yes	100.2 ± 20.78	21.8 ± 3.10	47.1 ± 9.41	8.8 ± 4.14	13.0 ± 4.94	9.5 ± 4.25
Experience of smoking cessation training after becoming a PHN						
No	95.6 ± 21.37 **	21.7 ± 3.20	44.6 ± 9.86 ***	8.3 ± 4.42	11.8 ± 5.16 ***	9.0 ± 4.27
Yes	102.0 ± 22.07 **	21.8 ± 3.31	48.1 ± 10.21 ***	8.9 ± 4.20	13.8 ± 5.11 ***	9.4 ± 4.38
Educational background						
Vocational college	101.6 ± 22.38	22.0 ± 3.19	47.7 ± 10.34	8.8 ± 4.72	13.9 ± 5.37 *	9.3 ± 4.46
Junior college	99.2 ± 22.71	21.7 ± 3.37	46.5 ± 10.47	8.6 ± 3.89	13.2 ± 5.36 *	9.2 ± 4.52
Above university	98.0 ± 21.56	21.7 ± 3.27	46.0 ± 10.05	8.6 ± 4.21	12.4 ± 5.04 *	9.2 ± 4.23

^a^ independent samples *t* test. ^b^ one-way analysis of variance (ANOVA). * *p* < 0.05; ** *p* < 0.01; *** *p* < 0.001.

**Table 3 ijerph-17-06135-t003:** Pearson correlation coefficients matrix of study variables (*n* = 521).

Items	5As	SE	KTSND	PDS	RUC	Age	Experience Years
5As		0.621 **	−0.113 *	0.331 **	0.413 **	0.125 **	0.136 **
SE			−0.109 *	0.369 **	0.418 **	0.256 **	0.274 **
KTSND				−0.037	0.014	−0.098 *	−0.136 **
PDS					0.483 **	0.068	0.089 *
RUC						0.043	0.071
Age							0.901 **
Experience years							

5As, ask, advise, assess, assist and arrange for follow-up; SE, self-efficacy scale; KTSND, kano test for social nicotine dependence; PDS, professional development scale; RUC, research utilization competency; * *p* < 0.05; ** *p* < 0.01.

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
