# Peer review of "Factors Associated with Prenatal Smoking Cessation Interventions among Public Health Nurses in Japan"

_ijerph, 2020, doi:10.3390/ijerph17176135_

Round 1
Reviewer 1 Report
The manuscript by Li et al reported the development of a potentially useful PSCI scale in Japan and some initial results supporting the scale’s internal consistency and construct validity. While the work is novel and significant to a considerable extent, I have found that the manuscript needs substantial improvements towards publication.
One major shortcoming of the study is that the validity assessments used such as smoking cessation intervention self-efficacy (SE) and research utilization competency (RUC) are not the most meaningful ones to definitely corroborate the construct of the PSCI, which was based on the 5A’s model. In comparison, the extent to which smokers who worked with the PHNs received the 5A intervention components could serve as a more relevant proof of the PSCI’s validity. Thus, the limitations of using SE and RUC for validity evaluation should be discussed.
Given the correlative nature of PSCI with SE and RUC, the interpretation presented in Lines 207-209 may not be fully justified. The relationship between PSCI and SE/RUC is likely to be reciprocal rather than unidirectional.
Other than the assessment of convergent validity, the study also lacks a demonstration of differential or divergent validity of the PSCI.
Another problem with the manuscript is the Figure 2 lacks clarity. First, a detailed caption should be provided to describe clearly the elements displayed. For instance, it is not clear why some values were in green while others in black. Beside item loading coefficients, what are those other values close to boxes representing the individual items? Second, many values in the figure are not legible.
It is unclear how the responders represented a national sample. While the authors reported the age distribution matched existing data, it would be informative to describe how the responders represented a national sample in other aspects such as geographical distribution.
In 3.2 Item Analysis, it was mentioned that 11 item showing ceiling effect and 1 item showing floor effect were retained. Then how is it possible to observe significant mean differences between the first and fourth quartiles for these items? Please clarify this.
In Lines 164-165, “The item with error correlation due to the similarity of question
expression was considered to draw the line of correlation with a model” seems poorly written. It seems that the authors were trying to say that, in step-wise model refitting, goodness-of-fit was improved by allowing estimation of covariance of the error terms between some items. The covariance presumably reflects the similarity in question presentation between those items. Please clarify this.
Language is yet another area where this manuscript can be significantly improved. Overall, the authors should carefully correct numerous problems with English grammar and expressions. I listed some, but not all, language issues (and probable fixes) as follows.
Line 14: Competency: competency.
Line 18: show: shows.
Line 39: 29: Twenty-nine.
Line 43: Whereas: In contrast (or comparison).
Line 65: PHN..: PHN.
Line 76: pregnant women, adults, elderly adults: inappropriate in-parallel listing a category with sub-categories.
Line 109: five stages, Firstly: five stages. Firstly
Line 135: 50: Fifty.
Line 163: good-of-fit: goodness-of-fit
Line 210: do not create: did not create.
Line 212: create: created.
Line 225: extracted: picked.
Author Response
Response to Reviewer 1 Comments
At first, we wish to express our deep appreciation to reviewer 1 for your insightful comments on our paper. We feel the comments have helped us significantly improve the paper.
Secondly, we are sorry that we give up the theme “Development of A Prenatal Smoking Cessation Intervention Scale for Public Health Nurses in Japan” based on our discussion and consideration. We think this study mainly discuss the factors associated with prenatal smoking cessation interventions based on the 5A’s model among Japanese PHNs. Therefore, we changed the theme to “Factors Associated with Prenatal Smoking Cessation Interventions among Public Health Nurses in Japan”, and rewrite this manuscript.
Thirdly, we answered these questions below.
The manuscript by Li et al reported the development of a potentially useful PSCI scale in Japan and some initial results supporting the scale’s internal consistency and construct validity. While the work is novel and significant to a considerable extent, I have found that the manuscript needs substantial improvements towards publication.
Point 1: One major shortcoming of the study is that the validity assessments used such as smoking cessation intervention self-efficacy (SE) and research utilization competency (RUC) are not the most meaningful ones to definitely corroborate the construct of the PSCI, which was based on the 5A’s model. In comparison, the extent to which smokers who worked with the PHNs received the 5A intervention components could serve as a more relevant proof of the PSCI’s validity. Thus, the limitations of using SE and RUC for validity evaluation should be discussed.
Given the correlative nature of PSCI with SE and RUC, the interpretation presented in Lines 207-209 may not be fully justified. The relationship between PSCI and SE/RUC is likely to be reciprocal rather than unidirectional.
Other than the assessment of convergent validity, the study also lacks a demonstration of differential or divergent validity of the PSCI.
Response 1: We really thank the reviewer for pointing out this very important issue. As the reviewer said, using SE and RUC for validity evaluation is limited and it should be further discussed. We think that SE and RUC should be discussed as the factors associated with prenatal smoking cessation interventions. Therefore, we changed the theme to “Factors Associated with Prenatal Smoking Cessation Interventions among Public Health Nurses in Japan”, and rewrite this manuscript.
Point 2: Another problem with the manuscript is the Figure 2 lacks clarity. First, a detailed caption should be provided to describe clearly the elements displayed. For instance, it is not clear why some values were in green while others in black. Beside item loading coefficients, what are those other values close to boxes representing the individual items? Second, many values in the figure are not legible.
Response 2: We really thank the reviewer for pointing out this very important issue. With the change of the theme, we deleted Figure 2 and gave up confirmatory factor analysis on PSCI scale.
Point 3: It is unclear how the responders represented a national sample. While the authors reported the age distribution matched existing data, it would be informative to describe how the responders represented a national sample in other aspects such as geographical distribution.
Response 3: We really thank the reviewer for pointing out this very important issue. We are sorry that we did not add the item “Work region” that was included in our questionnaire in the paper “Development of A Prenatal Smoking Cessation Intervention Scale for Public Health Nurses in Japan”. In the revised manuscript, we added the item “Work region” in Table 1 and explained it in Lines 265-267. The results of the item “Work region” were the following:
|
Table 1 Characteristics of respondents included in the analysis (n=521) |
||||
|
Characteristics |
n |
% |
Mean(±SD) |
|
|
Work region |
|
|
|
|
|
|
Hokkaido/Tohoku |
104 |
20.0 |
|
|
|
Kanto/Koshinetsu |
134 |
25.7 |
|
|
|
Tokai/Hokkiku |
107 |
20.5 |
|
|
|
Kinki |
71 |
13.6 |
|
|
|
Chugoku/Shikoku |
54 |
10.4 |
|
|
|
Kyushu/Okinawa |
51 |
9.8 |
|
We think that the geographical distribution may represent a national sample.
Point 4: In Lines 164-165, “The item with error correlation due to the similarity of question expression was considered to draw the line of correlation with a model” seems poorly written.
It seems that the authors were trying to say that, in step-wise model refitting, goodness-of-fit was improved by allowing estimation of covariance of the error terms between some items. The covariance presumably reflects the similarity in question presentation between those items. Please clarify this.
Response 4: We really thank the reviewer for pointing out this very important issue and revising the expression. With the change of the theme, we deleted Figure 2 and gave up confirmatory factor analysis on PSCI scale.
Point 5: Language is yet another area where this manuscript can be significantly improved. Overall, the authors should carefully correct numerous problems with English grammar and expressions. I listed some, but not all, language issues (and probable fixes) as follows.
Line 14: Competency: competency.
Line 18: show: shows.
Line 39: 29: Twenty-nine.
Line 43: Whereas: In contrast (or comparison).
Line 65: PHN..: PHN.
Line 76: pregnant women, adults, elderly adults: inappropriate in-parallel listing a category with sub-categories.
Line 109: five stages, Firstly: five stages. Firstly
Line 135: 50: Fifty.
Line 163: good-of-fit: goodness-of-fit
Line 210: do not create: did not create.
Line 212: create: created.
Line 225: extracted: picked.
Response 5: We really thank the reviewer for correcting the problems with English grammar and expressions. As the reviewer suggested, we have corrected these problems in the revised manuscript.
At last, we would like to thank the reviewer for useful suggestions that helped us to improve the original manuscript again. We give up the theme “Development of A Prenatal Smoking Cessation Intervention Scale for Public Health Nurses in Japan” and we think that it should be further discussed.

Reviewer 2 Report
Thank you for the opportunity to read your manuscript.
Here are my comments:
- Abstract: there were 1988 nurses invited to particpate, but only 554 provided data that were included in the analysis. Therefore, it is not correct to mention 1988 paricipants.
- Introduction: no comments
- Methods: Where is figure 1? I think it would clarify the methods section
- Results: Where are Supplementary file 1 and 2? Due to missing information, it is hard to comment on this section
- Discussion: this is not a discussion, but a repetion of the results. I suggest to rewrite this section.
Line 212: smoking cessation treatment drugs are not recommended: what with NRT (nicotine replacement therapy, which is allowed to use during pregnancy)?
Line 223: limitations: I suggest to elaborate on the low response rate. Did the researchers send a reminder? Or did they use other methods to increase the response rate? Was it not an option to collect data during a longer period? Can the researchers conclude this is an effective instrument? - References: there are no numbers in front of the references, so I had to count to see where you used which references. On p. 10 and 11 references are mentioned by the first name of the authors in stead of the last name
Author Response
Response to Reviewer 2 Comments
At first, we wish to express our deep appreciation to reviewer 2 for your insightful comments on our paper. We feel the comments have helped us significantly improve the paper.
Secondly, we are sorry that we give up the theme “Development of A Prenatal Smoking Cessation Intervention Scale for Public Health Nurses in Japan” based on our discussion and consideration. We think this study mainly discuss the factors associated with prenatal smoking cessation interventions based on the 5A’s model among Japanese PHNs. Therefore, we changed the theme to “Factors Associated with Prenatal Smoking Cessation Interventions among Public Health Nurses in Japan”, and rewrite this manuscript.
Thirdly, we answered these questions below.
Thank you for the opportunity to read your manuscript.
Here are my comments:
Point 1: Abstract: there were 1988 nurses invited to participate, but only 554 provided data that were included in the analysis. Therefore, it is not correct to mention 1988 participants.
Response 1: We really thank the reviewer for pointing out this very important issue. As the reviewer said, it is not correct to mention 1988 participants because only 521 in the revised manuscript provided data. We think that “The study subjects” is more appropriate for the expression. Therefore, we correct this sentence “Participants were 1988 PHNs working in 431 health centers of municipalities and special wards.” to the sentence “The study subjects were 1988 PHNs working in 431 health centers of municipalities and special wards across the country.”
Point 2: Introduction: no comments
Response 2: We really thank the reviewer for pointing out this very important issue. As the reviewer said, there are no comments in the “Introduction”. Therefore, we add two comments in the “Introduction” below.
Lines 52-53: “Therefore, it is an urgent public health concern to protect pregnant women from the harmful effects of tobacco.”
Lines 63-64: “Overall, psychological intervention should be prioritized to help women to stop smoking during pregnancy.”
Point 3: Methods: Where is figure 1? I think it would clarify the methods section
Response 3: We are really sorry that we failed to upload the figure 1 because of our mistakes. With the change of the theme, we replaced figure 1 with the Supplementary file 1 that was attached in the revised manuscript.
In addition, we rewrite and clarify the methods section.
Point 4: Results: Where are Supplementary file 1 and 2? Due to missing information, it is hard to comment on this section
Response 4: We are really sorry that the Supplementary file 1 and 2 were not uploaded because of the mistake from Assistant Editor of IJERPH. We have received the email from Assistant Editor, where Assistant Editor hope we can give you an explanation about this mistake. In the revised manuscript, we have Supplementary file 1 and 2 which were well attached.
Point 5: Discussion: this is not a discussion, but a repetion of the results. I suggest to rewrite this section.
Line 212: smoking cessation treatment drugs are not recommended: what with NRT (nicotine replacement therapy, which is allowed to use during pregnancy)?
Response 5: We really thank the reviewer for pointing out this very important issue. As the reviewer suggested, we have rewrite this section.
Line 223: limitations: I suggest to elaborate on the low response rate. Did the researchers send a reminder? Or did they use other methods to increase the response rate? Was it not an option to collect data during a longer period? Can the researchers conclude this is an effective instrument?
Response 5: We really thank the reviewer for pointing out this very important issue. We sent a reminder about one month after the questionnaire was handed out. However, the response rate was still very low (26.2%).
We are sorry that we did not set a longer period to collect the questionnaire (only about two mouth).
We are sorry that we did not conclude this is an effective instrument. We just think that sending a reminder about one month after the questionnaire was handed out might be useful for increasing the response rate.
Point 6: References: there are no numbers in front of the references, so I had to count to see where you used which references. On p. 10 and 11 references are mentioned by the first name of the authors instead of the last name
Response 6: We really thank the reviewer for pointing out this two important issues. We have added the numbers in front of the references in the revised manuscript. In addition, we have adjusted the first name of the authors instead of the last name.
At last, we would like to thank the reviewer for useful suggestions that helped us to improve the original manuscript again. We give up the theme “Development of A Prenatal Smoking Cessation Intervention Scale for Public Health Nurses in Japan” and we think that it should be further discussed.

Reviewer 3 Report
The manuscript “Development of A Prenatal Smoking Cessation Intervention Scale for Public Health Nurses in Japan” addresses an interesting topic.
General Concerns
- It was difficult to understand what authors wanted to measure: To delivery of 5As steps is performed, or, what is the level of implemented 5As carried out? The results shown do not meet the aims of “quantify” prenatal smoking cessation interventions.
- It is necessary to clarify if the PHN in Japan have been trained to implement a brief smoking cessation counseling intervention, the 5As: ask, advise, assess, assist, arrange. In some countries, not all nurses have been trained and -assist, arrange- is not part of their roles. This is important, it may cause "Non-response error".
- Authors should provide a context on smoking cessation in health centers of municipalities, e.g. stop smoking services (physician or PHN), NRT use and cost treatment.
- The number in the references was not this reviewers copy.
- Difference data: In abstract Line 17: GFI=0.938, AGFI=0.903, 17 CFI=0.938 and RMSEA=0.074. In results L 167: GFI was 0.850, AGFI was 0.819, CFI was 0.889, and RMSEA was 0.066. A CFI >0.95 and RMSEA <0.06 are indicative of a good model fit. Tip, consider dividing the sample to compare the results
Minor Points
L 12: PHNs, write full name on first occurrence in text
L 23: keywords are in health Sciences Descriptors?
L 191-198: This paragraph repeats the results and does not contribute to the discussion
L 296 – 298: Repeated reference
Author Response
Response to Reviewer 3 Comments
At first, we wish to express our deep appreciation to reviewer 3 for your insightful comments on our paper. We feel the comments have helped us significantly improve the paper.
Secondly, we are sorry that we give up the theme “Development of A Prenatal Smoking Cessation Intervention Scale for Public Health Nurses in Japan” based on our discussion and consideration. We think this study mainly discuss the factors associated with prenatal smoking cessation interventions based on the 5A’s model among Japanese PHNs. Therefore, we changed the theme to “Factors Associated with Prenatal Smoking Cessation Interventions among Public Health Nurses in Japan”, and rewrite this manuscript.
Thirdly, we answered these questions below.
The manuscript “Development of A Prenatal Smoking Cessation Intervention Scale for Public Health Nurses in Japan” addresses an interesting topic.
General Concerns
Point 1: It was difficult to understand what authors wanted to measure: To delivery of 5As steps is performed, or, what is the level of implemented 5As carried out? The results shown do not meet the aims of “quantify” prenatal smoking cessation interventions.
Response 1: We really thank the reviewer for pointing out this very important issue. As the reviewer said, the results shown do not meet the aims of “quantify” prenatal smoking cessation interventions. Therefore, we changed the theme to “Factors Associated with Prenatal Smoking Cessation Interventions among Public Health Nurses in Japan”, and rewrite this manuscript.
In addition, if we keep on doing scale development, is it better to quantify prenatal smoking cessation intervention competence?
Point 2: It is necessary to clarify if the PHN in Japan have been trained to implement a brief smoking cessation counselling intervention, the 5As: ask, advise, assess, assist, arrange. In some countries, not all nurses have been trained and -assist, arrange- is not part of their roles. This is important, it may cause "Non-response error".
Response 2: We really thank the reviewer for pointing out this very important issue. As the reviewer said, if nurses have been not trained to implement the 5As in some countries, it may cause "Non-response error". However, this study subjects were Japanese PHNs who were expected to implement the 5As in Japanese guidelines for smoking cessation. As a limitation of this study, we wrote the “PHNs in the sample are from Japan, thus the results may not be applicable to PHNs in other countries.” in Lines 310-311 of the revised manuscript.
Point 3: Authors should provide a context on smoking cessation in health centers of municipalities, e.g. stop smoking services (physician or PHN), NRT use and cost treatment.
Response 3: We really thank the reviewer for pointing out this very important issue. Smoking cessation treatment service was started and covered by the Japanese medical insurance system since 2006. Briefly, smoking cessation treatment service consists of a total of five sessions: first session and then follow up sessions at 2, 4, 8 and 12 weeks after the first session. At each session, the current smoking behavior is confirmed by exhaled carbon monoxide (CO) concentration and the smoker’s self-reported smoking status. Meanwhile, patients receive medication treatment consisting of varenicline (standard use: 12 weeks) or nicotine patches (standard use: 8 weeks), and behavioral counselling from physicians and nurses.
However, in Japan, smoking cessation treatment service was only provided in smoking cessation outpatients or clinics, and it was not provided in health centers. PHNs in health centers can only provide behavioral counselling for special people such as pregnant women, child abuse family using the chance of home visit or health center interview. In addition, the behavioral counselling training was provided for medical worker such as PHNs by “Japan Smoking Cessation Training Outreach Project”. (http://www.nosmoke-med.org/jstop)
Therefore, we did not provide a context on smoking cessation stop smoking services (physician or PHN), NRT use and cost treatment in health centers of municipalities, because PHN only provide behavioral counselling.
Point 4: The number in the references was not this reviewers copy.
Response 4: We really thank the reviewer for pointing out this very important issue. We have added the numbers in front of the references in the revised manuscript.
Point 5: Difference data: In abstract Line 17: GFI=0.938, AGFI=0.903, 17 CFI=0.938 and RMSEA=0.074. In results L 167: GFI was 0.850, AGFI was 0.819, CFI was 0.889, and RMSEA was 0.066. A CFI >0.95 and RMSEA <0.06 are indicative of a good model fit. Tip, consider dividing the sample to compare the results
Response 5: We really thank the reviewer for pointing out this difference data in Lines 17 and 167. With the change of the theme, we deleted the result of goodness-of-fit and gave up confirmatory factor analysis on PSCI scale.
Minor Points
L 12: PHNs, write full name on first occurrence in text
L 23: keywords are in health Sciences Descriptors?
L 191-198: This paragraph repeats the results and does not contribute to the discussion
L 296 – 298: Repeated reference
Response 6: We really thank the reviewer for pointing out these problems.As the reviewer suggested, we have corrected these problems in the revised manuscript.
At last, we would like to thank the reviewer for useful suggestions that helped us to improve the original manuscript again. We give up the theme “Development of A Prenatal Smoking Cessation Intervention Scale for Public Health Nurses in Japan” and we think that it should be further discussed.

Round 2
Reviewer 2 Report
Thank you for the extensive work on the revision.
All documents are added and clarify the manuscript.
Title: is an improvement, much clearer.
Introduction: by adding the 3 sentences, the introduction is adapted to the new title and is clear.
Methods: a big improvement! Also thanks to the supplementary files, this section is easier to understand and clearer.
Line 142: "Smoking cessation intervention factors include the experience of 142 smoking cessation training in school days and after becoming a PHN." Isn't there a better word for 'school days'? For me, it has a connotation with secondary school, but I think you mean training to become a PHN?
Results: also a big improvement! Added tables clarify the manuscript.
Discussion:
Line 231 to 256: it is a logical conclusion that older PHN who use the 5 A's have more working experience. Maybe these 2 paragraphs can be integrated?
Suggestion: you send a reminder after 1 month in order to increase the response rate (response 5), maybe you can mention this effort?
References: number 31 and 32: this is the same author Hein de Vries, so his name should be mentioned in the same way: de Vries, ...
Author Response
Response to Reviewer 2 Comments
At first, we wish to express our deep appreciation to reviewer 2 for your insightful comments on our paper again. We really feel the comments have helped us significantly improve the paper.
Secondly, we have made the following modifications based on your comments.
Thank you for the extensive work on the revision.
All documents are added and clarify the manuscript.
Title: is an improvement, much clearer.
Introduction: by adding the 3 sentences, the introduction is adapted to the new title and is clear.
Methods: a big improvement! Also thanks to the supplementary files, this section is easier to understand and clearer.
Point 1: Line 142: "Smoking cessation intervention factors include the experience of 142 smoking cessation training in school days and after becoming a PHN." Isn't there a better word for 'school days'? For me, it has a connotation with secondary school, but I think you mean training to become a PHN?
Response 1: We really thank the reviewer for pointing out this very important issue. As the reviewer said, the 'school days' really has a connotation with secondary school, whereas I mean the training vocational college or university to become a PHN. Therefore, I think the 'college days' is a better word to express the mean based on the explanation from the Oxford Dictionary.
College: (in the UK) a place where students go to study or to receive training after they have left school.
(in the US) a university where students can study for a degree after they have left school.
University: an institution at the highest level of education where you can study for a degree or do research.
School: a place where children go to be educated.
With the change of this word, we also changed other places where 'school days' is used.
Results: also a big improvement! Added tables clarify the manuscript.
Discussion:
Point 2: Line 231 to 256: it is a logical conclusion that older PHN who use the 5 A's have more working experience. Maybe these 2 paragraphs can be integrated?
Response 2: We really thank the reviewer for pointing out this very important issue. In the previous studies and this study, “age” and “working experience” are set as two independent items. It may imply that the association between age and experience years cannot be indiscreetly asserted. However, as the reviewer said, the logical conclusion can be considered. Therefore, we added the following sentence in Lines 264 to 266 of the revised manuscript.
Lines 264 to 266: …Furthermore, the above two results imply that mature nurses may have more work experience to use the 5A's and then the experience motivates them to implement the 5A's, although these studies do not provide the evidence on the association between age and experience years.
Point 3: Suggestion: you send a reminder after 1 month in order to increase the response rate (response 5), maybe you can mention this effort?
Response 3: We really thank the reviewer for pointing out this very important issue. As the reviewer suggested, we added the following sentence in Lines 112 to 113 of the revised manuscript.
Lines 112 to 113: …To increase the response rate, we sent a reminder about one month after the questionnaire was mailed out. …
Point 4: References: number 31 and 32: this is the same author Hein de Vries, so his name should be mentioned in the same way: de Vries, ...
Response 4: We are sorry for this mistake. We have already modified it.
At last, we would like to thank the reviewer for useful suggestions that helped us to improve the original manuscript again.
Reviewer 3 Report
The manuscript “Factors Associated with Prenatal Smoking Cessation 2 Interventions among Public Health Nurses in Japan” was a new article.
General Concerns
- Introduction, include all relevant references (factors associated)
- Discussion, I suggest explaining why “work regions” are positively associated. Also, why the Kano test for social nicotine dependence (KTSND) was negatively associated with 5A. Organize discussion according to motivational factors, abilities factors and barriers, as reported in methods to make the discussion clearer and make more precise conclusions.
- In conclusion, according to the results discussed: authors should give a recommendation for strengthening the health services. What factors to consider to improve cessation services for the population.
- Improve table 2, use of “]” is not clear
Minor Points
L 160: After point, writing capital letter.
Author Response
Response to Reviewer 3 Comments
At first, we wish to express our deep appreciation to reviewer 3 for your insightful comments on our paper again. We really feel the comments have helped us significantly improve the paper.
Secondly, we have made the following modifications based on your comments.
The manuscript “Factors Associated with Prenatal Smoking Cessation Interventions among Public Health Nurses in Japan” was a new article.
General Concerns
Introduction, include all relevant references (factors associated)
Point 1: Discussion, I suggest explaining why “work regions” are positively associated.
Response 1: We really thank the reviewer for pointing out this very important issue. We are sorry that we don’t find the related evidence on why PHNs’ “work regions” are positively associated with the 5A’s.
In Japan, there are several smoking cessation training programs such as Japan Smoking Cessation Training Outreach Project (J-STOP), smoking cessation support workshop for health professionals sponsored by national cancer center that provide the smoking-related knowledge, behavioral counselling and medication treatment on smoking cessation. We think that the work region where more PHNs complete the training program may be positively associated with the 5A’s. However, we don’t find the related paper on the status of receiving the training program among Japanese PHNs in different work region.
Therefore, we can only provide the fact and implication in Lines 282 to 290 of the revised manuscript and in below.
Fourthly, our study also revealed that PHNs who were from Chugoku/Shikoku and were less likely to implement the 5A’s, however PHNs who were from Kinki and Tokai/Hokkiku were more likely to implement the 5A’s. In Japan, there are several smoking cessation training programs such as Japan Smoking Cessation Training Outreach Project (J-STOP), smoking cessation support workshop for health professionals sponsored by national cancer center that provide the smoking-related knowledge, behavioral counselling and medication treatment on smoking cessation. We think that the work region where more PHNs complete the training program may be positively associated with the 5A’s. Therefore, this consideration may imply that it is necessary to promote more PHNs who were from Chugoku/Shikoku to complete the training program.
Point 2: Also, why the Kano test for social nicotine dependence (KTSND) was negatively associated with 5A.
Response 2: We really thank the reviewer for pointing out this very important issue. We have already written this section on why the Kano test for social nicotine dependence (KTSND) was negatively associated with 5A in Lines 299 to 309 of the revised manuscript and in below.
…Regarding the association between the KTSND and the 5A’s, several previous studies reported that smokers had more significant social nicotine dependence than non-smokers and ex-smokers [44,62-64] and smokers with higher KTSND scores was very difficult to start to quit smoking among the stages for quitting smoking [44]. Another previous study reported that non-smoking and ex-smoking teachers with high KTSND scores might underestimate the health hazards of second-hand smoke, and overestimate the recommended protective measures [65]. In this study, we found that the KTSND was negatively associated with the 5A’s. This result is almost consistent with one previous study which reported that participants on the Annual Meeting of the Japan Lung Cancer Society who answered that they are not interested in smoking cessation intervention showed significantly higher KTSND scores [64]. Overall, health workers with significant social nicotine dependence were less likely to implement the smoking cessation interventions. …
Point 3: Organize discussion according to motivational factors, abilities factors and barriers, as reported in methods to make the discussion clearer and make more precise conclusions.
In conclusion, according to the results discussed: authors should give a recommendation for strengthening the health services. What factors to consider to improve cessation services for the population.
Response 3: We really thank the reviewer for pointing out this very important issue. We have already written this section on giving a recommendation for strengthening the health services in Lines 312 to 316 and in below.
In the future, there are two suggestions for the prenatal smoking cessation intervention services. On the one hand, continuing smoking cessation training should be widely implemented to strengthen the intervention competence among PHNs. On the other hand, self-efficacy for smoking cessation support should be further strengthened by promoting research utilization competency, regulating the health professionals’ norms and improving the attitudes on smoking.
Point 4: Improve table 2, use of “]” is not clear
Response 4: We really thank the reviewer for pointing out this issue. We have improved table 2 in the revised manuscript.
Point 5: Minor Points
L 160: After point, writing capital letter.
Response 5: We really thank the reviewer for pointing out this point. We have revised this point.
At last, we would like to thank the reviewer for useful suggestions that helped us to improve the original manuscript again.
